# Joint Shapley values: a measure of joint feature importance

**Chris Harris**
Tokyo, Japan
Raptor Financial Technologies
chrisharriscjh@gmail.com

**Richard Pymar**
Economics, Mathematics & Statistics
Birkbeck College University of London, UK
r.pymar@bbk.ac.uk

**Colin Rowat**
Economics
University of Birmingham, UK
c.rowat@bham.ac.uk

## Abstract

The Shapley value is one of the most widely used measures of feature importance partly as it measures a feature's average effect on a model's prediction. We introduce joint Shapley values, which directly extend Shapley's axioms and intuitions: joint Shapley values measure a set of features' average contribution to a model's prediction. We prove the uniqueness of joint Shapley values, for any order of explanation. Results for games show that joint Shapley values present different insights from existing interaction indices, which assess the effect of a feature within a set of features. The joint Shapley values provide intuitive results in ML attribution problems. With binary features, we present a presence-adjusted global value that is more consistent with local intuitions than the usual approach.

## 1 Introduction

*Game theory*'s Shapley value partitions the value arising from joint efforts among individual agents (Shapley, 1953). Specifically, denote by $N = \{1, \ldots, n\}$ a set of agents, and by $\mathcal{G}^N$ the set of games on $N$, where a game is a set function $v$ from $2^N$ to $\mathbb{R}$ with $v(\varnothing) = 0$. Then $v(S)$ is the worth created by coalition $S \subseteq N$. When there is no risk of confusion, we omit braces to indicate singletons (e.g. $i$ rather than $\{i\}$) and denote a set's cardinality by the corresponding lower case letter (e.g. $s = |S|$).

For any agent $i$, Shapley's value is then

$$\psi_i(v) \equiv \sum_{S \subseteq N \setminus \{i\}} \frac{s!\,(n-s-1)!}{n!} \left[ v\left(S \cup i\right) - v\left(S\right) \right]. \tag{1}$$

This is the average worth that $i$ adds to possible coalitions $S$, weighted as follows: if the set of agents, $S$, has already 'arrived', draw the next agent, $i$, to arrive uniformly over the remaining $N \setminus S$ agents.

Shapley's value is widely used in *explainable AI*'s *attribution problem*, partitioning model predictions among individual features (q.v. Štrumbelj & Kononenko (2014); Lundberg & Lee (2017)) after the model has been trained. Evaluating the prediction function at a specific feature value corresponds to an agent's presence; evaluating it at a reference (or baseline) feature value corresponds to the agent's absence. A feature's Shapley value is its average marginal contribution to the model's predictions.

When features are correlated, individual measures of importance may mislead (Bhatt et al., 2020; Patel et al., 2021), as measures of individual significance such as the $t$-test do. Thus, Shapley's value has been extended to sets of features (Grabisch & Roubens, 1999a; Marichal et al., 2007; Alshebli et al., 2019; Dhamdhere et al., 2020). As these extensions introduce axioms not present in Shapley, they do not preserve the Shapley value's intuition.

We extend Shapley's axioms to sets of features, randomizing over sets rather than individual features. The resulting joint Shapley value thus directly extends Shapley's value to a measure of sets of feature importance: the average marginal contribution of a set of features to a model's predictions.

Our approach's novelty is seen in our extension of the null axiom: in Shapley (1953), a null agent contributes nothing to any set of agents to which it may belong; here, a null set contributes nothing to any set to which it may belong. By contrast, *interaction indices* (Grabisch & Roubens, 1999a; Alshebli et al., 2019; Dhamdhere et al., 2020) recursively decompose sets into individual elements, retaining the original Shapley null axiom. Thus, these indices measure sets' contributions relative to their constituent elements — so are complementary to the joint Shapley value.

The generalised Shapley value (Marichal et al., 2007) is closer to our work, but differs in a number of respects: our probabilities are independent of the size of the set of features under consideration[1]; our efficiency axiom is fully joint, while theirs are based on singletons and pairs; our symmetry axiom provides uniqueness without reliance on recursion or partnership axioms.

To illustrate, consider a null coalition, $T$, whose individual members contribute positively to coalitions that they join. Interaction indices assign a negative value to $T$, as the individual members act discordantly together. However, from a joint feature importance viewpoint, $T$ should be assigned value 0. The joint Shapley value matches this intuition.

In the movie review application presented below, the joint Shapley values reveal contributions of collections of words, including grammatical features such as negation ({disappointed} versus {won't, disappointed}), and adjectives ({effort} versus {terrific, effort}).

Like the Shapley-Taylor interaction index (Dhamdhere et al., 2020), our efficiency axiom depends on a positive integer $k$, the *order of explanation*, which limits the number of joint Shapley values to those for subsets up to cardinality $k$. As there are $2^N - 1$ non-empty subsets of $N$, the full set of joint Shapley values rapidly becomes unmanageable otherwise. In practice, $k$ should be set to trade off insight (favouring higher $k$) against computational cost (favouring lower $k$).

For each $k$, the extended axioms yield a unique joint Shapley value. Unlike in Shapley (1953), the joint anonymity and symmetry axioms are not interchangeable: each imposes distinct restrictions.

Section 2 presents and extends the original Shapley axioms. Section 3 introduces joint Shapley values, deriving them as the unique solution to the extended axioms. Section 4 illustrates joint Shapley values in the game theoretical environment and applies them to the Boston housing (Harrison & Rubinfeld, 1978) and Rotten Tomatoes movie review (Pang & Lee, 2005) datasets, comparing them to interaction indices and presenting a sampling technique to facilitate calculation. Section 5 concludes. The Appendix collects proofs and other supplemental material.

## 2    EXTENDING SHAPLEY'S AXIOMS

For a game $v \in \mathcal{G}^N$, and a permutation $\sigma$ on $N$, denote a *permuted game* by $\sigma v \in \mathcal{G}^N$ such that $\sigma v(\sigma(S)) = v(S), \forall S \subseteq N$, where $\sigma(S) = \{\sigma(i) : i \in S\}$. An index $\phi(v)$ of the game $v \in \mathcal{G}^N$ is any real-valued function on $2^N$.

The original axioms uniquely satisfied by the (standard) Shapley value $\psi$, are:

**LI** *linearity* : $\psi$ is a linear function on $\mathcal{G}^N$, i.e. $\psi(v + w) = \psi(v) + \psi(w)$ and $\psi(av) = a\psi(v)$ for any $v, w \in \mathcal{G}^N$ and $a \in \mathbb{R}$.

**NU** *null* : An agent that adds no worth to any coalition has no value, i.e. if $v(S \cup \{i\}) = v(S)$ for all $S \subseteq N \setminus \{i\}$, then $\psi_i(v) = 0$. This axiom is sometimes called *dummy*.

**EF** *efficiency* : The sum of the values of all agents is equal to the worth of the entire set, i.e. for all $v \in \mathcal{G}^N$, $\sum_{i=1}^n \psi_i(v) = v(N)$.

**AN** *anonymity* : For any $\sigma$ on $N$ and any $v \in \mathcal{G}^N$, $\psi_i(v) = \psi_{\sigma(i)}(\sigma v)$, for all $i \in N$.

---

[1]Measures of joint significance (e.g. $F$-tests) and model selection (e.g. BIC or AIC) penalize larger feature sets to avoid overfitting. As joint Shapley values are calculated after model training, this problem does not arise.

**SY** *symmetry* : If two agents add equal worth to all coalitions that they can both join then they receive equal value: if $v(S \cup \{i\}) = v(S \cup \{j\}) \forall S \subseteq N \setminus \{i, j\}$ then $\psi_i(v) = \psi_j(v)$. This is strictly weaker than anonymity.

Now extend each of these axioms in natural ways to conditions on sets rather than singletons. Below, $\phi_S(v)$ denotes an index for coalition $S$ on game $v$.

**JLI** *joint linearity* : $\phi$ is a linear function on $\mathcal{G}^N$, i.e. $\phi(v + w) = \phi(v) + \phi(w)$ and $\phi(av) = a\phi(v)$ for any $v, w \in \mathcal{G}^N$ and $a \in \mathbb{R}$. (This axiom has not been modified.)

**JNU** *joint null* : A *coalition* that adds no worth to any coalition has no value, i.e. if $v(S \cup T) = v(S)$ for all $S \subseteq N \setminus T$, then $\phi_T(v) = 0$.

**JEF** *joint efficiency* : The sum of the values of all *coalitions* up to cardinality $k$ is equal to the worth of the entire set, i.e. for all $v \in \mathcal{G}^N$,

$$\sum_{\substack{\varnothing \neq T \subseteq N: \\ |T| \leq k}} \phi_T(v) = v(N).$$

**JAN** *joint anonymity* : For any $\sigma$ on $N$ and any $v \in \mathcal{G}^N$, $\phi_T(v) = \phi_{\sigma(T)}(\sigma v)$, for all $T \subseteq N$.

**JSY** *joint symmetry* : If two *coalitions* perform equally when joining coalitions that they can both join *and for other coalitions they add no worth* then they receive an equal value, i.e. if

1. $v(S \cup T) = v(S \cup T')$ for all $S \subseteq N \setminus (T \cup T')$,
2. $v(S \cup T) = v(S)$ for all $S \subseteq N \setminus T$ such that $S \cap T' \neq \varnothing$,
3. $v(S \cup T') = v(S)$ for all $S \subseteq N \setminus T'$ such that $S \cap T \neq \varnothing$,

then $\phi_T(v) = \phi_{T'}(v)$.

Axiom JSY only equates the joint Shapley values for coalitions $T$ and $T'$ if they contribute identically to coalitions that they may both join, and contribute nothing to the other coalitions. Axioms JLI, JEF and JAN are all also used in Dhamdhere et al. (2020). Our joint null and joint symmetry notions appear to be new: they reflect our interest in a set of features' contribution to a model's predictions, so that the set's cardinality should not play a role in determining its value.

## 3 Joint Shapley values

Our main result is that there is a unique solution to axioms JLI, JNU, JEF, JAN and JSY, the joint Shapley value. The uniqueness is up to the $k^{\text{th}}$ order of explanation; we say nothing about $|T| > k$.

**Theorem 1.** *For each order of explanation $k \in \{1, \ldots, n\}$, there is a unique (up to the $k^{\text{th}}$ order of explanation) index $\phi^J$ which satisfies axioms JLI, JNU, JEF, JAN and JSY. It has the form*

$$\phi_T^J(v) = \sum_{S \subseteq N \setminus T} q_{|S|}[v(S \cup T) - v(S)]$$

*for each $\varnothing \neq T \subseteq N$ with $|T| \leq k$, where $(q_0, \ldots, q_{n-1})$ uniquely solves the recursive system*

$$q_0 = \frac{1}{\sum_{i=1}^{k} \binom{n}{i}}, \quad q_r = \frac{\sum_{s=(r-k) \vee 0}^{r-1} \binom{r}{s} q_s}{\sum_{s=1}^{k \wedge (n-r)} \binom{n-r}{s}}; \tag{2}$$

*for all $r \in 1, \ldots, n - 1$.*

When $k = 1$, the joint Shapley values coincide with Shapley's values.[2] When $k = n$, we have

$$q_r = \sum_{j=0}^{r} \binom{r}{j} \frac{(-2)^{r-j}}{2^{n-j} - 1}, \quad \forall r \in \{0, \ldots, n-1\}.$$

For each $k$, the constants $q_r$ are non-negative and satisfy $\sum_{s=n-k}^{n-1} \binom{n}{s} q_s = 1$. Further, as the joint Shapley value is similar in form to equation (1)'s standard Shapley value, the value of coalition $T$

---

[2]The Shapley-Taylor interaction index, $\phi^{ST}$, also has this property (Dhamdhere et al., 2020).

depends on its marginal contribution to other coalitions; unlike interaction indices, it does not depend on the worth of its constituent agents.

As with the Shapley value, the joint Shapley value can be seen as the worth brought by 'arriving' agents but, rather than arriving one at a time, they can now also arrive in coalitions. We develop this interpretation in Appendix A.

We show here the implication of each of the joint axioms introduced above and prove Theorem 1.

It is already known that joint linearity restricts measures to be linear combinations of worths:

**Lemma 1** (Grabisch & Roubens (1999a), Proposition 1). *If $\phi$ satisfies JLI, then for every $\varnothing \neq T \subseteq N$ there exists a family of real constants $\{a_T^S\}_{S \subseteq N}$ such that for every $v \in \mathcal{G}^N$,*

$$\phi_T(v) = \sum_{S \subseteq N} a_S^T v(S).$$

Axiom JNU then constrains the values of the constants $\{a_S^T\}$:

**Lemma 2.** *Suppose $\phi$ satisfies JLI and JNU and let $\{a_T^S\}_{S \subseteq N, \varnothing \neq T \subseteq N}$ be the constants from Lemma 1. Then for every $\varnothing \neq T \subseteq N$ and $\varnothing \neq S \subseteq N \setminus T$, $a_S^T = -a_{S \cup T}^T$. Further, for every $\varnothing \neq T \subseteq N$, $S \subseteq N \setminus T$, and $\varnothing \neq H \subsetneq T$, $a_{S \cup H}^T = 0$.*

SCombining these two lemmas yields:

**Proposition 1.** *Suppose $\phi$ satisfies JLI and JNU. Then there exist constants $\{p^T(S)\}$ that depend on $T$ and $S$ such that for every $\varnothing \neq T \subseteq N$ and $v \in \mathcal{G}^N$*

$$\phi_T(v) = \sum_{S \subseteq N \setminus T} p^T(S)[v(S \cup T) - v(S)]. \tag{3}$$

Now establish a condition on the $\{p^T(S)\}$ values which must be satisfied under JEF:

**Proposition 2.** *For each order of explanation $k$, $\phi$ satisfies axioms JLI, JNU and JEF if and only if for every $\varnothing \neq T \subseteq N$ with $|T| \leq k$ and $v \in \mathcal{G}^N$, $\phi_T(v) = \sum_{S \subseteq N \setminus T} p^T(S)[v(S \cup T) - v(S)]$ with $\{p^T(S)\}$ satisfying*

$$\delta_N(S) = \sum_{\substack{\varnothing \neq T \subseteq S: \\ |T| \leq k}} p^T(S \setminus T) - \sum_{\substack{\varnothing \neq T \subseteq N \setminus S: \\ |T| \leq k}} p^T(S), \tag{4}$$

*for all $\varnothing \neq S \subseteq N$, where $\delta_N(S)$ equals 1 if $S = N$ and 0 otherwise.*

Recall that symmetry axiom SY is strictly weaker than anonymity axiom AN (Malawski, 2020). This is not the case for their joint counterparts: each imposes a different constraint on the constants $\{p^T(S)\}$, as we shall see here. First, consider the effect of imposing JAN.

**Proposition 3.** *For each order of explanation $k$, $\phi$ satisfies axioms JLI, JNU, JEF and JAN if and only if for every $\varnothing \neq T \subseteq N$ and $v \in \mathcal{G}^N$, $\phi_T(v) = \sum_{S \subseteq N \setminus T} p^T(S)[v(S \cup T) - v(S)]$ with $\{p^T(S)\}$ satisfying (4) and*

$$p^T(S) = p^{T'}(S') \, \forall \varnothing \neq T, T' \subseteq N, \, S \subseteq N \setminus T, \, S' \subseteq N \setminus T' \text{ s.t. } s = s', t = t'. \tag{5}$$

The analogous result for JSY is:

**Proposition 4.** *For each order of explanation $k$, $\phi$ satisfies axioms JLI, JNU, JEF and JSY if and only if for every $\varnothing \neq T \subseteq N$ and $v \in \mathcal{G}^N$, $\phi_T(v) = \sum_{S \subseteq N \setminus T} p^T(S)[v(S \cup T) - v(S)]$ with $\{p^T(S)\}$ satisfying (4) and*

$$p^T(S) = p^{T'}(S) \, \forall \varnothing \neq T, T' \subseteq N, \, S \subseteq N \setminus (T \cup T'). \tag{6}$$

Combining Propositions 2–4 completes the proof of Theorem 1.

The computational complexity of deriving the $(q_0, \ldots, q_{n-1})$ is $\mathcal{O}(nk^2)$. Once these have been determined, the joint Shapley values can be calculated; the complexity of doing so is $\mathcal{O}(3^n \wedge (2^n n^k))$.

# 4 EXPERIMENTS

## 4.1 GAME THEORETICAL

We present two game theoretical models from Dhamdhere et al. (2020) with known 'ground truths'. For each, we compare the joint Shapley value, the Shapley interaction index (Grabisch & Roubens, 1999b), the generalised Shapley value (Marichal et al., 2007), the added-value index (Alshebli et al., 2019), and the Shapley-Taylor interaction index (Dhamdhere et al., 2020), respectively:

$$\phi_T^{SI}(v) \equiv \sum_{S \subseteq N \setminus T} \frac{1}{n-t+1} \binom{n-t}{s}^{-1} \sum_{L \subseteq T} (-1)^{t-l} v(S \cup L), \forall T \subseteq N;$$

$$\phi_T^{GS}(v) \equiv \sum_{S \subseteq N \setminus T} \frac{(n-s-t)!s!}{(n-t+1)!} [v(S \cup T) - v(S)];$$

$$\phi_T^{AV}(v) \equiv v(T) - \sum_{i \in T} \frac{1}{2^{n-1}} \sum_{S \subseteq N : i \in S} \sum_{C \subseteq S \setminus i} \frac{c!(s-c-1)!}{s!} [v(C \cup i) - v(C)];$$

$$\phi_T^{ST}(v;k) \equiv \begin{cases} \sum_{L \subseteq T} (-1)^{t-l} v(L) & \text{if } t < k \\ \frac{k}{n} \sum_{S \subseteq N \setminus T} \binom{n-1}{s}^{-1} \sum_{L \subseteq T} (-1)^{t-l} v(S \cup L) & \text{if } t = k. \end{cases}$$

Table 1: Joint and interaction measures in $n = 3$ game theory examples

| $T$ | $v$ | Shapley $(k=1)$ | $\phi^{SI}$ | $\phi^{GS}$ | $\phi^{AV}$ | $\phi^{ST}(v_m;k)$ $k=2$ | $k=3$ | $\phi^J(v_m;k)$ $k=2$ | $k=3$ |
|---|---|---|---|---|---|---|---|---|---|
| | | The majority game: $v_m(T) = 1$ when $t \geq 2$, and is equal to 0 otherwise | | | | | | | |
| $i$ | 0 | $1/3$ | $1/3$ | $1/3$ | $-1/3$ | 0 | 0 | $1/9$ | $2/21$ |
| $i,j$ | 1 | 0 | 0 | $1/2$ | $1/3$ | $1/3$ | 1 | $2/9$ | $4/21$ |
| $N$ | 1 | | $-2$ | 1 | 0 | | $-2$ | | $3/21$ |
| | | A linear model with crosses: $v_c(T) = \sum_{i \in T} 1 + c \max\{0, t-2\}$ for $c \in \mathbb{R}$ | | | | | | | |
| $i$ | 1 | $1/3(3+c)$ | $1/3(3+c)$ | $1/3(3+c)$ | $-1/12c$ | 1 | 1 | $5/18(2+c)$ | $5/21(2+c)$ |
| $i,j$ | 2 | | $1/3c$ | $1/2(4+c)$ | $-1/6c$ | $1/3c$ | 0 | $1/18(8+c)$ | $1/21(8+c)$ |
| $N$ | $3+c$ | | $c$ | $3+c$ | $3/4c$ | | $c$ | | $3/21(3+c)$ |

In the $n = 3$ majority game, Table 1 shows that, while Shapley-Taylor assigns 0 to singletons, the joint Shapley value recognises that singletons contribute positively when joining existing coalitions. The Shapley-Taylor and Shapley Interaction indices assign negative values to $N$ (for discordant interaction between members); the joint Shapley rewards $N$ for adding worth in its own right.

In the linear model with crosses, joint Shapley values' signs can again differ from those of interaction indices: let $c = -2$, which sets $\phi_1^J = 0$: if $i = 1$ joins the empty coalition or either singleton, it adds unit worth; however, when $i = 1$ joins $\{2, 3\}$, it subtracts unit worth. Averaging (according to the arrival order), gives a net contribution of 0: coalition 1 does not contribute any worth in expectation. By contrast, the Shapley-Taylor value always (for $k > 1$) assigns value to singletons equal to their worth, as it is not (by design) capturing information about expected contributions of features.

## 4.2 THE AI/ML ATTRIBUTION PROBLEM

Following Štrumbelj & Kononenko (2010), let $f$ be a prediction function, and $\boldsymbol{x} = (x_1, \ldots, x_n) \in \mathcal{A}$ an instance from the feature space. For a set of features $S \subseteq N$, define the prediction difference $v_{\boldsymbol{x}}(S)$ when only features in $S$ are known, as

$$v_{\boldsymbol{x}}(S) \equiv \frac{1}{|\mathcal{A}|} \sum_{\boldsymbol{z} \in \mathcal{A}} [f(\tau(\boldsymbol{x}, \boldsymbol{z}, S)) - f(\boldsymbol{z})], \tag{7}$$

where $\tau(\boldsymbol{x}, \boldsymbol{z}, S) \in \mathcal{A}$ is defined as $\tau(\boldsymbol{x}, \boldsymbol{z}, S)_i$ equal to $x_i$ for $i \in S$ and $z_i$ otherwise. Thus $v_{\boldsymbol{x}}(S)$ is the difference between the expected prediction when only the feature values of $\boldsymbol{x}$ in $S$ are known,

and the expected prediction when no feature values are known. Then for each $T \subseteq N$, $\phi_T^J(v_{\boldsymbol{x}})$ is the contribution of features $T$ to the prediction $f(\boldsymbol{x})$. To estimate $\phi_T^J(v_{\boldsymbol{x}})$, let $X$ be a random variable whose law coincides with the law of the number of agents already present when $T$ arrives in the arrival interpretation (so that the law of $X$ depends on $t$, the size of $T$):

$$\mathbb{P}(X = i) = \binom{n-t}{i} q_i \left( \sum_{j=0}^{n-t} \binom{n-t}{j} q_j \right)^{-1}$$

for each $i \in \{0, \ldots, n\}$ and $\mathcal{S}$ be a random variable uniform on the set of subsets of $N \setminus T$ of size $X$. Once $X$ is sampled, we can generate the set of agents already arrived, called $S$, by choosing it uniformly from all subsets of $N \setminus T$ of size precisely $X$. Then

$$\phi_T^J(v_{\boldsymbol{x}}) = \sum_{j=0}^{n-t} \binom{n-t}{j} q_j \mathbb{E}[v_{\boldsymbol{x}}(T \cup \mathcal{S}) - v_{\boldsymbol{x}}(\mathcal{S})].$$

Thus, by the law of large numbers, we can estimate this expectation by taking an average of $v_i(T \cup S_i) - v_i(S_i)$ where $S_i$ has the same distribution as $\mathcal{S}$ and $v_i(S) = f(\tau(\boldsymbol{x}, z_i, S)) - f(z_i)$, where $z_i$ is chosen uniformly from the feature space. Refer to $\phi_T^J(v_{\boldsymbol{x}})$ as the *local* joint Shapley value of features $T$ at instance $\boldsymbol{x}$. We can combine local values to obtain, for each feature, a *global* joint Shapley value. We do so in two ways. The first is the standard methodology (q.v. Lundberg & Lee (2017)), which averages the absolute values of locals. We introduce the second for models with exclusively binary variables by considering presence/absence of features $T$ in $\boldsymbol{x}$. This *presence-adjusted* global joint Shapley value is defined as

$$\tilde{\phi}_T^J(f) = \frac{1}{|\mathcal{A}|} \sum_{\boldsymbol{x} \in \mathcal{A}} (2 \cdot \mathbf{1}_{\boldsymbol{x}(T)=1} - 1) \phi_T^J(v_x), \tag{8}$$

where $\mathbf{1}_{\boldsymbol{x}(T)=1}$ is one if all features $T$ are present in $\boldsymbol{x}$, and 0 otherwise. Covert et al. (2020) introduced SAGE (Shapley Additive Global Importance), a more sophisticated treatment of global influence measures that maintains efficiency at the global level.

All experiments are run on a single Intel(R) Core(TM) i7-6820HQ CPU. Training details and tuning parameters are provided in the accompanying code.

### 4.2.1 SIMULATED DATA

Consider three features, $x_1, x_2$ and $x_3$, each uniformly drawn from $(0, 1)$ and a dataset of 50 observations. We first consider independently drawn features; we then investigate correlation by fixing $x_2 = 1 - x_1$. We use simplified 'ML models', $f(\boldsymbol{x})$, to obtain exact global joint Shapley values, averaging the absolute values of local joint Shapley values derived from equation (7). Table 2 displays the results.

Table 2: Uniform random variables; $k = 3$

|  | $x_1$ | $x_2$ | $x_3$ | $x_1, x_2$ | $x_1, x_3$ | $x_2, x_3$ | $x_1, x_2, x_3$ |
|---|---|---|---|---|---|---|---|
| | independent features | | | | | | |
| $f_1(\boldsymbol{x}) = x_1$ | 0.122 | 0 | 0 | 0.049 | 0.049 | 0 | 0.037 |
| $f_2(\boldsymbol{x}) = x_1 + x_2$ | 0.122 | 0.114 | 0 | 0.061 | 0.049 | 0.045 | 0.046 |
| $f_3(\boldsymbol{x}) = x_1 - x_2$ | 0.122 | 0.114 | 0 | 0.062 | 0.049 | 0.045 | 0.047 |
| | correlated features: $x_2 = 1 - x_1$ | | | | | | |
| $f_2(\boldsymbol{x})$ | 0.122 | 0.122 | 0 | 0 | 0.049 | 0.049 | 0 |
| $f_3(\boldsymbol{x})$ | 0.122 | 0.122 | 0 | 0.098 | 0.049 | 0.049 | 0.073 |

**Independent variables** As only $x_1$ influences $f_1$, only coalitions including it receive value; the more diluted its role, the lower the value; $x_2$ and $x_3$ are symmetrically irrelevant. For $f_2$, $x_1$ and $x_2$ play equal roles, and receive equal values (up to variance due to the sample size). For $f_2$ and $f_3$, $\{x_1, x_2\}$ receives similar value, and more than in $f_1$, where only $x_1$ influenced model predictions. Comparing $f_2$ and $f_3$ shows that values only differ for coalitions containing both $x_1$ and $x_2$.

**Correlated features** $f_2$ now exhibits what we term a *cancellation effect* between $x_1$ and $x_2$, assigning a value of zero to coalitions containing both. Similarly, $f_3$ reveals an *enhancement effect*: values assigned to coalitions with both $x_1$ and $x_2$ are larger than in the independent case.

Enhancement and cancellation effects do not uniquely identify underlying phenomena. To illustrate, let $x_1, x_2$ and $x_3$ be independent Bernoulli(0.5) random variables. Setting $k = 3$ and letting $n \to \infty$, we obtain the exact presence-adjusted global joint Shapleys in Table 3.

Table 3: Bernoulli(0.5) random variables; $k = 3$

| $f(\boldsymbol{x})$ | $x_1$ | $x_2$ | $x_3$ | $x_1, x_2$ | $x_1, x_3$ | $x_2, x_3$ | $x_1, x_2, x_3$ |
|---|---|---|---|---|---|---|---|
| $f_1(\boldsymbol{x}) = x_1$ | $5/21$ | $0$ | $0$ | $1/21$ | $1/21$ | $0$ | $1/56$ |
| $f_2(\boldsymbol{x}) = x_1 + x_2$ | $5/21$ | $5/21$ | $0$ | $2/21$ | $1/21$ | $1/21$ | $1/28$ |
| $f_4(\boldsymbol{x}) = x_1 x_2$ | $5/42$ | $5/42$ | $0$ | $1/14$ | $1/42$ | $1/42$ | $3/112$ |

The joint Shapley values containing $x_1$ and $x_2$ are larger for models $f_2$ and $f_4$ than for model $f_1$, a different sort of enhancement effect from that above. However, the joint Shapley value for $x_1$ is smaller in $f_4$ than it is in $f_1$ or $f_2$, again a different type of cancellation effect.

Finally, joint Shapley values can provide insight into the black-box model's structure. If, for example, $f$ can be decomposed as $f(\boldsymbol{x}) = g(x_1) + h(x_2, \ldots, x_n)$ with independent Bernoulli(0.5) random variables, then the presence-adjusted global joint Shapley value of $x_1$ is $1/2 \left(g(1) - g(0)\right) \sum_{s=0}^{n-1} \binom{n-1}{s} q_s = 5/21$, as shown in the table. If the joint Shapley value deviates from this, we reject the decomposition, as in the case of $f_4$.

### 4.2.2 BOSTON HOUSING DATA

For comparability, we follow Dhamdhere et al. (2020) by training a random forest on the Boston housing dataset (Harrison & Rubinfeld, 1978), computing global values using the first notion discussed above. Table 4 presents the largest and smallest global joint Shapley and Shapley-Taylor values.

Table 4: Joint Shapley values for the Boston dataset

| Shapley | | $\phi^{ST}(f; k)$ | | $\phi^J(f; k)$ | |
|---|---|---|---|---|---|
| $(k = 1)$ | $k = 2$ | $k = 3$ | | $k = 2$ | $k = 3$ |
| RM: 2.57 | RM: 3.12 | RM: 3.12 | | LSTAT: 0.63 | LSTAT: 0.42 |
| LSTAT: 2.47 | LSTAT: 2.04 | LSTAT: 2.04 | | RM: 0.56 | RM: 0.34 |
| AGE: 0.81 | DIS: 1.55 | DIS: 1.55 | | LSTAT, RM: 0.30 | AGE: 0.11 |
| DIS: 0.63 | CRIM: 1.37 | CRIM: 1.37 | | AGE, LSTAT: 0.21 | LSTAT, RM: 0.11 |
| CRIM: 0.46 | B: 1.31 | DIS, LSTAT: 1.33 | | AGE, RM: 0.20 | NOX: 0.10 |
| NOX: 0.44 | NOX: 1.15 | B: 1.31 | | DIS, RM: 0.19 | DIS: 0.08 |
| PTRATIO: 0.27 | ⋮ | ⋮ | | ⋮ | ⋮ |
| B: 0.24 | CHAS, RM: 0.15 | AGE, CRIM, RM: 0.21 | | CHAS, TAX: 0.02 | RAD, TAX, ZN: 0.00 |
| TAX: 0.22 | LSTAT, TAX: 0.15 | AGE, DIS, PTRATIO: 0.21 | | RAD, ZN: 0.01 | CHAS: 0.00 |
| INDUS: 0.19 | INDUS, RAD: 0.15 | DIS, LSTAT, PTRATIO: 0.21 | | CHAS, RAD: 0.01 | CHAS, RAD, TAX: 0.00 |
| RAD: 0.13 | NOX, TAX: 0.14 | AGE, LSTAT, PTRATIO: 0.20 | | ZN: 0.01 | CHAS, RAD, ZN: 0.00 |
| ZN: 0.07 | DIS, INDUS: 0.13 | AGE, NOX, RM: 0.20 | | CHAS: 0.01 | CHAS, TAX, ZN: 0.00 |
| CHAS: 0.07 | DIS, PTRATIO: 0.12 | B, CRIM, LSTAT: 0.19 | | CHAS, ZN: 0.01 | CHAS, ZN: 0.00 |

For $k = 1$, the joint Shapley values are the classical Shapley values. The unimportance of CHAS seems to reflect its low variance (only 35 of 506 units back onto the Charles).

For $k = 2$, LSTAT and RM still have the largest joint Shapley values, with {LSTAT, RM} having third largest. The top pairs all involve LSTAT or RM, indicating that these variables also contribute to explaining house prices jointly with other variables. By contrast, {LSTAT, RM} is the 16th largest Shapley-Taylor interaction value. The $k = 2$ singleton joint Shapley values are fairly evenly distributed throughout the pairs; by contrast, the singleton $k = 2$ Shapley-Taylor values outrank all the pairs. This is consistent with our extension of axiom NU to not favour singletons. The largest joint Shapley value involving NOX is {NOX, LSTAT}, which is about five times as large as {NOX, DIS}

and {NOX, RAD}. This is consistent with Harrison and Rubinfeld's observation that NOX offset RAD and DIS, but reinforced LSTAT: knowing the values of NOX and LSTAT adds a lot of predictive power; knowing the values of NOX and RAD or DIS tends to wash out additional predictive power.

For $k = 3$, singletons again dominate: the largest triple is {AGE, LSTAT, RM}, the three individually most important features. Continuing the above analysis, the joint Shapley value of {DIS, RAD} is about 0.015, while that of {DIS, NOX, RAD} is about 0.008, and that of {DIS, LSTAT, RAD} is about 0.022. We understand this to mean {DIS, NOX, RAD} contains the same sort of information as {DIS, RAD}, but with NOX offsetting the other variables, while LSTAT brings novel socio-economic information to the distance variables DIS and RAD, which it continues to partially offset. Similarly, the least important joint features include the individually unimportant CHAS and its variants —- echoing the role played by RM and LSTAT at the top of the list. While {DIS, LSTAT} has a Shapley-Taylor value of 1.33 (indicating a large interaction between the pair but not its overall importance), its joint Shapley is 0.06 (ranked 18th) — directly assigning a value of importance.

Table 4's values are exact. Figure 1 demonstrates the estimation procedure above, showing convergence of the sampled values as the iterations increase. As with other sampling procedures (e.g. Štrumbelj & Kononenko (2014)), this cuts the complexity to order $n$ to a linear power of $k$; as $k \rightarrow n$, the complexity approaches $2^n$ times a polynomial in $n$ (compared to $3^n$ for the exact $\phi^J$).

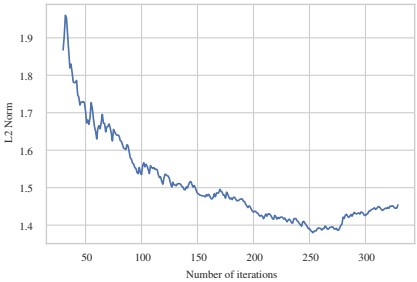

Figure 1: Sampled $\phi^J$ converges to exact $\phi^J$ with $L^2$ norm $\approx 1.45$ ; $k = 2$ Boston

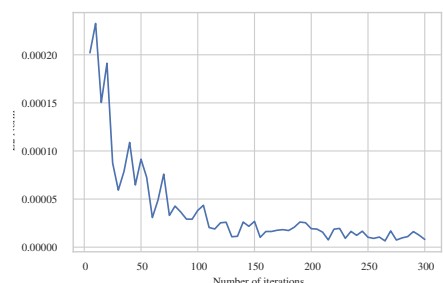

Figure 2: Difference between consecutive $\phi^J$ samples averages converges to zero; $k = 2$ movie review #2

### 4.2.3 MOVIE REVIEWS

We train a fully connected neural network (two hidden layers, 16 units per layer, ReLU activations) on the binary movie review classifications in Pang & Lee (2005). From the full set of reviews, we remove a test block of 100 (picked to include those in Table 1 of Dhamdhere et al. (2020)) for analysis. We encode reviews as the 1000 most common words in the corpus, augmented by key words in Table 1 of Dhamdhere et al. (2020) to aid comparison of measures, for a total of 1004. (The DSA features are drawn from positive reviews: {won't}, {disappointed}, {both}, {inspiring}, {a}, {crisp}, {excellent}, {youthful}, {John}, {terrific}.) Binary accuracy is typically c. 76% after four epochs.

Table 5's local joint Shapley values are tiny as each of the $2^{1004} - 1$ joint features has a tiny effect on the probability of a positive review. In the vein of Dhamdhere et al., we identify intuitive effects:

1. *negation*: the negative local joint Shapley values for {disappointed} and {be, disappointed} become positive when negated by adding {won't} to the coalition.
2. *enhancement*: the positive local joint Shapley values for {both} and {and} are enhanced by the positive {and, both}.
3. *context*: the sign of local joint Shapley values involving {well} depend on the context; while {you, well} is positive, {left, well} is negative.
4. *lost potential*: the positive local joint Shapley values involving {fascinating} become negative when {been} is added.
5. *adjectival*: while {director} and {effort} are individually negative, they become positive when qualified with the adjectives {winning} and {terrific}, respectively.

Table 5: Examples of local joint Shapley values in the Pang & Lee (2005) movie reviews

| Review | joint Shapleys |
|---|---|
| 1: *negation*: aficionados of the whodunit won't be disappointed | {disappointed}: $-2 \times 10^{-5}$
{won't}: $6 \times 10^{-5}$
{be, disappointed}: $-9 \times 10^{-8}$
{won't, disappointed}: $6 \times 10^{-8}$
{won't, be, disappointed}: $5 \times 10^{-9}$ |
| 2: *enhancement*: both inspiring and pure joy | {both}: $2 \times 10^{-4}$
{and}: $6 \times 10^{-5}$
{and, both}: $1 \times 10^{-6}$ |
| 3: *context*: you wish Jacquot had left well enough alone | {you, well}: $9 \times 10^{-7}$
{left, well}: $-3 \times 10^{-7}$ |
| 4: *lost potential*: fascinating little thriller that would have been perfect | {would}: $-1 \times 10^{-4}$
{fascinating}: $2 \times 10^{-4}$
{would, fascinating}: $3 \times 10^{-7}$
{would, been, fascinating}: $-1 \times 10^{-8}$ |
| 5: *adjectival*: director …award-winning …make a terrific effort | {effort}: $-1 \times 10^{-5}$
{director}: $-9 \times 10^{-6}$
{terrific, effort}: $8 \times 10^{-7}$
{winning, director}: $5 \times 10^{-7}$ |

For global values, as the features are binary, we compute the presence-adjusted global joint Shapleys, $\tilde{\phi}^J$. Comparing the DSA features' global $\tilde{\phi}^J$ across the test reviews for $k = 1$ and $k = 2$ shows {terrific} to be the largest, followed by {both} then {excellent}. This reflects prevalence in the training set, where {terrific} appears four times as often in positive reviews. Similarly, on the negative $\tilde{\phi}^J$ side, {John} is followed by {disappointed}: {John} appears thrice as often in negative reviews.

The largest pairs are {both, inspiring} followed by {excellent, youthful}. All of the pairs, except {John, terrific}, have a positive sign: these pairs typically do not appear outside the positive reviews from which they were drawn. As our encoding discards sequential information, the pairs are smaller than the singletons: a coalition merely indicates co-occurrence in a review, rather than a bigram.

As the 10 DSA features are a very small subset of all 1004, we cannot asses whether JEF holds. However, as a health check note: the average presence-adjusted local joint Shapleys over the positive reviews is about 20% larger than that over the negative reviews. They are both negative: as these features are drawn from positive reviews, but largely absent in any given review, their effect is negative. Further, Table 6 indicates that the ranking of single features is largely preserved as $k$ increases from 1 to 2: the exception is {a, crisp}: in the corpus, {crisp} is typically preceded by {a}.

Figure 2 shows the estimated values' convergence.

Table 6: Largest presence-adjusted global joint Shapley values on DSA features

| $k = 1$ | terrific | both | excellent | won't | crisp | youthful | a | inspiring | disappointed | John |
|---|---|---|---|---|---|---|---|---|---|---|
| $k = 2$ | terrific | both | excellent | won't | a | youthful | inspiring | crisp | disappointed | John |

## 5  CONCLUSIONS

The joint Shapley value directly extends Shapley's value to measure the effect of a set of features on a model's predictions. Further work is needed to maintain properties like global efficiency (Covert et al., 2020) and to make sampling more efficient (Williamson & Feng, 2020; Mitchell et al., 2021). We believe that understanding complex models is labor intensive. Nevertheless, we envisage a common workflow that computes $k = 1$ Shapley values, followed by $k = 2$ to identify strong pairwise effects; analyses for $k \geq 3$ then respond to the analyst's evolving questions about the model's functioning.

## 6 ETHICS STATEMENT

This paper contributes to the literature on explainable AI, an important component of the Fairness, Accountability and Transparency research agenda for ethical AI. It does not use human subjects; it only draws on publicly available datasets; the work has not been sponsored, and does not seek to promote any third organisations; none of the authors face any conflict of interest.

## 7 REPRODUCIBILITY STATEMENT

Our proofs and source code are available in the accompanying supplemental material; all data are taken from the public domain.

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

In this supplementary material we discuss the arrival order interpretation, present the proofs of all results, and discuss a notion of joint symmetry obtained by removing conditions 2 and 3 from JSY.

## A    ARRIVAL ORDER INTERPRETATION

As discussed in the paper, the joint Shapley value can be viewed in terms of the worth brought by 'arriving' agents but, rather than arriving one at time, they can now also arrive in coalitions. To be precise, consider this procedure: at time 0, no agents have arrived; at each $t \in \{1, 2, \ldots\}$, the next set of agents to arrive is chosen uniformly from the set of non-empty subsets of size at most $k$ of the remaining (yet to arrive) agents. Then $\phi_T^J$ is the expected worth brought by coalition $T$ when it arrives (a coalition is assigned zero worth if it does not arrive at any time). To see this, denote by $A_i$ the coalition to arrive at time $i$, by $B_i$ the union of all coalitions that have arrived up to time $i$: $B_i = \bigcup_{j \le i} A_j$, and by $p_T$ the probability that at some time coalition $T$ arrives, $p_T = \mathbb{P}(\exists\, i : B_i = T)$. We have the recursive relationship:

$$
p_T = \sum_{i=1}^{n} \sum_{\substack{S \subseteq T: \\ |S| \ge |T| - k}} \mathbb{P}(B_{i-1} = S)\mathbb{P}(A_i = T \setminus S \mid B_{i-1} = S)
$$

$$
= \sum_{i=1}^{n} \sum_{\substack{S \subsetneq T: \\ |S| \ge |T| - k}} \mathbb{P}(B_{i-1} = S) \left( \sum_{r=1}^{(n-|S|)\wedge k} \binom{n-|S|}{r} \right)^{-1} = \sum_{s=(|T|-k)\vee 0}^{|T|-1} \binom{t}{s} p_S \left( \sum_{r=1}^{(n-s)\wedge k} \binom{n-s}{r} \right)^{-1},
$$

where in the last line $S$ is any set of size $s$. Thus we see that $p_T$ only depends on $T$ through its cardinality, and defining

$$
\hat{p}_t := p_T \left( \sum_{r=1}^{(n-t)\wedge k} \binom{n-t}{r} \right)^{-1}
$$

for any $T$ with $|T| = t$, the expected worth brought by coalition $T$ under this procedure is

$$
\sum_{i=1}^{n} \sum_{S \subseteq N \setminus T} \mathbb{P}(B_{i-1} = S,\, A_i = T)[v(S \cup T) - v(S)] = \sum_{S \subseteq N \setminus T} \hat{p}_{|S|}[v(S \cup T) - v(S)].
$$

Further, we have the relationship

$$
\hat{p}_t = \frac{\sum_{s=(t-k)\vee 0}^{t-1} \binom{t}{s} p_s}{\sum_{r=1}^{(n-t)\wedge k} \binom{n-t}{r}}
$$

and since we know that $p_0 = 1$ (we start with no agents having arrived) and $p_n = 1$ (we finish with all agents having arrived) we also have the identities:

$$
1 = \sum_{s=n-k}^{n-1} \binom{n}{s} \hat{p}_s, \quad 1 = \hat{p}_0 \sum_{r=1}^{k} \binom{n}{s}.
$$

By comparing with the identities defining $(q_0, \ldots, q_{n-1})$, we deduce that $\hat{p}_t = q_t$ for all $t \in \{0, \ldots, n-1\}$, which verifies this arrival interpretation.

## B    PROOFS

*Proof of Lemma 2.* For the first statement, for each $\varnothing \neq S \subseteq N \setminus T$ consider the game

$$v_S(R) = \begin{cases} 1 & \text{if } R = S \text{ or } R = S \cup T, \\ 0 & \text{otherwise.} \end{cases}$$

Then for such $S$, by JNU, $\phi_T(v_S) = 0$. By this equality, Lemma 1, and the definition of $v_S$,

$$0 = \phi_T(v_S) = \sum_{R \subseteq N} a_R^T v_S(R) = a_S^T + a_{S \cup T}^T.$$

For the second statement, for each $\varnothing \neq H \subsetneq T$ let $\alpha_H$ be a constant and for $S \subseteq N \setminus T$, consider the game

$$x_S^{\boldsymbol{\alpha}}(R) = \begin{cases} \alpha_H & \text{if } R = S \cup H \text{ for some } \varnothing \neq H \subsetneq T, \\ 0 & \text{otherwise.} \end{cases}$$

By JNU, $\phi_T(x_S^{\boldsymbol{\alpha}}) = 0$ for every $S \subseteq N \setminus T$. Thus by Lemma 1

$$0 = \phi_T(x_S^{\boldsymbol{\alpha}}) = \sum_{R \subseteq N} a_R^T x_S^{\boldsymbol{\alpha}} = \sum_{\varnothing \neq H \subsetneq T} a_{S \cup H}^T x_S^{\boldsymbol{\alpha}}(S \cup H) = \sum_{\varnothing \neq H \subsetneq T} a_{S \cup H}^T \alpha_H.$$

Since this holds for every choice of constants $\alpha_H$, it follows that $a_{S \cup H}^T = 0$ for all $S \subseteq N \setminus T$ and $\varnothing \neq H \subsetneq T$, as required.    $\square$

*Proof of Proposition 1.* By Lemma 1 there exist constants $\{a_S^T\}_{S \subseteq N, \varnothing \neq T \subseteq N}$ such that for every $v$ and $\varnothing \neq T \subseteq N$,

$$\phi_T(v) = \sum_{S \subseteq N} a_S^T v(S) = \sum_{S \subseteq N \setminus T} \left( a_S^T v(S) + \sum_{\varnothing \neq H \subsetneq T} a_{S \cup H}^T v(S \cup H) + a_{S \cup T}^T v(S \cup T) \right)$$

$$= \sum_{S \subseteq N \setminus T} (a_S^T v(S) + a_{S \cup T}^T v(S \cup T)) = \sum_{S \subseteq N \setminus T} a_{S \cup T}^T [v(S \cup T) - v(S)];$$

where the last two equalities owe to Lemma 2. The proof is complete by setting $p^T(S) = a_{S \cup T}^T$.    $\square$

*Proof of Proposition 2.* Suppose $\phi$ satisfies axioms JLI, JNU and JEF. Then by Proposition 1 the constants $\{p^T(S)\}$ exist, such that for every $v$ and $\varnothing \neq T \subseteq N$,

$$\phi_T(v) = \sum_{S \subseteq N \setminus T} p^T(S)[v(S \cup T) - v(S)].$$

Now for each $\varnothing \neq R \subseteq N$ consider the identity game

$$w_R(S) = \begin{cases} 1 & \text{if } S = R, \\ 0 & \text{otherwise.} \end{cases}$$

Then for every $\varnothing \neq T \subseteq N$ with $|T| \leq k$,

$$\phi_T(w_R) = \sum_{S \subseteq N \setminus T} p^T(S)[w_R(S \cup T) - w_R(S)].$$

Note that the term $w_R(S \cup T) - w_R(S)$ in the above sum is equal to 1 only when $S \subsetneq R$ and $T = R \setminus S$, i.e. only when $S = R \setminus T$ and $\varnothing \neq T \subseteq R$. Further note that this term is equal to $-1$ only when $S = R$ and $T \neq \varnothing$, i.e. when $S = R$ and $\varnothing \neq T \subseteq N \setminus R$ (as must have $S \cap T = \varnothing$). In all other cases, this term is 0. Hence we deduce from JEF that

$$\delta_N(R) = w_R(N) = \sum_{\substack{\varnothing \neq T \subseteq R: \\ |T| \leq k}} p^T(R \setminus T) - \sum_{\substack{\varnothing \neq T \subseteq N \setminus R: \\ |T| \leq k}} p^T(R).$$

Now we show the implication in the other direction. If $\phi_T(v) = \sum_{S \subseteq N \setminus T} p^T(S)[v(S \cup T) - v(S)]$ then it is immediate that JLI and JNU are satisfied. For JEF, we wish to show that for every $v$,

$$\sum_{\substack{\varnothing \neq T \subseteq N: \\ |T| \leq k}} \sum_{S \subseteq N \setminus T} p^T(S)[v(S \cup T) - v(S)] = v(N).$$

Note that for each $\varnothing \neq R \subseteq N$, the coefficient of $v(R)$ on the left-hand side in the above equation is

$$\sum_{\substack{\varnothing \neq T \subseteq R: \\ |T| \leq k}} p^T(R \setminus T) - \sum_{\substack{\varnothing \neq T \subseteq N \setminus R: \\ |T| \leq k}} p^T(R).$$

But by equation (4), this is equal to $\delta_N(R)$. $\hfill\square$

*Proof of Proposition 3.* In light of Proposition 2 we just have to consider JAN.

**Only if:** Suppose $\phi$ satisfies JLI, JNU, JEF and JAN. First, we shall establish that

$$p^T(S) = p^T(S') \ \forall \varnothing \neq T \subseteq N, \ S, S' \subseteq N \setminus T \text{ s.t. } s = s'. \tag{9}$$

Fix such a $T$, $S$ and $S'$. Consider again the identity game, $w_S$ and let $\sigma$ be a self-inverse permutation such that $S \mapsto S'$, $S' \mapsto S$, and $\sigma(\{i\}) = \{i\}$ for all $i \notin S \cup S'$. As $T \subseteq N \setminus (S \cup S')$ and $\sigma(T) = T$, we have by JAN

$$\phi_T(w_S) = \phi_{\sigma^{-1}(T)}(w_S) = \phi_T(\sigma w_S)$$

where

$$\sigma w_S(R) = w_S(\sigma^{-1}(R)) = \left\{ \begin{array}{ll} 1 & \text{if } R = \sigma(S) = S' \\ 0 & \text{otherwise} \end{array} \right\} = w_{S'}(R).$$

Hence we obtain $\phi_T(w_S) = \phi_T(\sigma w_S) = \phi_T(w_{S'})$. Next, from Proposition 1 we have

$$\phi_T(w_S) = \sum_{Q \subseteq N \setminus T} p^T(Q)[w_S(Q \cup T) - w_S(Q)] = -p^T(S),$$

and similarly $\phi_T(w_{S'}) = -p^T(S')$. Hence we obtain $p^T(S) = p^T(S')$, showing (9).

Using induction on $s$, we now establish that (5) holds. Fix $T$ and $T'$ of the same size. For the base case, suppose $s = s' = n - t$. For $S \subseteq N \setminus T$ and $S' \subseteq N \setminus T'$, this forces $S = N \setminus T$ and $S' = N \setminus T'$. Now consider the game

$$x_n(R) = \begin{cases} 1 & \text{if } r = n \\ 0 & \text{otherwise,} \end{cases}$$

so that $x_n(R) = 1$ if and only if $R = N$. Define a self-inverse permutation $\sigma$ so that $\sigma(T) = T'$, $\sigma(T') = T$ and $\sigma(\{i\}) = \{i\}$ for all $i \notin (T \cup T')$. Then by JAN and as $\sigma x_n = x_n$,

$$\phi_T(x_n) = \phi_{\sigma^{-1}(T')}(x_n) = \phi_{T'}(\sigma x_n) = \phi_{T'}(x_n).$$

Next, from Proposition 1,

$$\phi_T(x_n) = \sum_{Q \subseteq N \setminus T} p^T(Q)[x_n(T \cup Q) - x_n(Q)] = p^T(N \setminus T),$$

and similarly $\phi_{T'}(w_n) = p^{T'}(N \setminus T')$. Hence we obtain $p^T(N \setminus T) = p^{T'}(N \setminus T')$ which establishes the base case.

We now suppose that $p^T(S) = p^{T'}(S')$ for all $s = s' \geq n - c$ where $S \subseteq N \setminus T$, $S' \subseteq N \setminus T'$ and $c$ is a positive integer. We shall show that $p^T(S) = p^{T'}(S')$ for all $s = s' \geq n - c - 1$ where where $S \subseteq N \setminus T$ and $S' \subseteq N \setminus T'$. To this end, consider the game

$$x(R) = \begin{cases} 1 & \text{if } r \geq n - c - 1 + t, \\ 0 & \text{otherwise} \end{cases}.$$

Thus, as before, we may write

$$\phi_T(x) = \sum_{Q \subseteq N \setminus T} p^T(Q)\left[x(Q \cup T) - x(Q)\right] = \sum_{\substack{Q \subseteq N \setminus T \\ n-c-1 \leq q < n-c-1+t}} p^T(Q).$$

Similarly,

$$\phi_{T'}(x) = \sum_{\substack{Q' \subseteq N \setminus T' \\ n-c-1 \leq q' < n-c-1+t}} p^T(Q).$$

Again, by JAN, we prove that $\phi_T(x) = \phi_{T'}(x)$. Define a self-inverse permutation $\sigma$ so that $\sigma(T) = T'$, $\sigma(T') = T$ and $\sigma(\{i\}) = \{i\}$ for all $i \notin (T \cup T')$. As worth in game $x$ depends only on a coalition's cardinality, we have $\sigma x = x$. Thus, by JAN, $\phi_T(x) = \phi_{\sigma(T)}(\sigma x) = \phi_{\sigma(T)}(x) = \phi_{T'}(x)$.

However,

$$\phi_T(x) = \sum_{\substack{Q \subseteq N \setminus T \\ q=n-c-1}} p^T(Q) + \sum_{\substack{Q \subseteq N \setminus T \\ n-c-1 < q < n-c-1+t}} p^T(Q) = \sum_{\substack{Q \subseteq N \setminus T \\ q=n-c-1}} p^T(Q) + \sum_{\substack{Q' \subseteq N \setminus T' \\ n-c-1 < q' < n-c-1+t}} p^{T'}(Q');$$

and

$$\phi_{T'}(x) = \sum_{\substack{Q' \subseteq N \setminus T' \\ q'=n-c-1}} p^{T'}(Q') + \sum_{\substack{Q' \subseteq N \setminus T' \\ n-c-1 < q' < n-c-1+t}} p^{T'}(Q')$$

which gives, by the inductive hypothesis and $\phi_T(x) = \phi_{T'}(x)$,

$$\sum_{\substack{Q \subseteq N \setminus T \\ q=n-c-1}} p^T(Q) = \sum_{\substack{Q' \subseteq N \setminus T' \\ q'=n-c-1}} p^{T'}(Q').$$

But by (9), $p^T(Q) = p^T(Q')$ if $q = q'$. Thus the above equation becomes

$$\binom{n-t}{n-c-1} p^T(Q) = \binom{n-t}{n-c-1} p^{T'}(Q')$$

for any $Q \subseteq N \setminus T$ and $Q' \subseteq N \setminus T'$ with $q = q' = n-c-1$. Thus, $p^T(Q) = p^{T'}(Q')$, completing the inductive step, and the 'only if' statement.

**If:** Suppose (5) is satisfied. Fix a permutation $\sigma$ on $N$ and game $v \in \mathcal{G}^N$. Then for any $\varnothing \neq T \subseteq N$,

$$\phi_T(\sigma v) = \sum_{S \subseteq N \setminus T} p^T(S)\left[\sigma v(S \cup T) - \sigma v(S)\right] = \sum_{S \subseteq N \setminus T} p^T(S)\left[v\left(\sigma^{-1}(S \cup T)\right) - v\left(\sigma^{-1}(S)\right)\right]$$

$$= \sum_{S \subseteq N \setminus T} p^T(S)\left[v\left(\sigma^{-1}(S) \cup \sigma^{-1}(T)\right) - v\left(\sigma^{-1}(S)\right)\right].$$

Defining the set $S' = \sigma^{-1}(S)$ allows us to rewrite the above as

$$\phi_T(\sigma v) = \cdots = \sum_{S' \subseteq N \setminus \sigma^{-1}(T)} p^T(\sigma(S'))\left[v\left(S' \cup \sigma^{-1}(T)\right) - v(S')\right]$$

$$= \sum_{S' \subseteq N \setminus \sigma^{-1}(T)} p^{\sigma^{-1}(T)}(S')\left[v\left(S' \cup \sigma^{-1}(T)\right) - v(S')\right] = \phi_{\sigma^{-1}(T)}(v),$$

with the penultimate step due to condition (5). $\qquad\square$

*Proof of Proposition 4.* In light of Proposition 2 we just have to consider JSY.

**Only if:** Suppose $\phi$ satisfies JLI, JNU, JEF and JSY, fix $\varnothing \neq T, T' \subseteq N$, and consider again the identity game $w_R$. Then for any $\varnothing \neq R \subseteq N \setminus (T \cup T')$,

- $w_R(S \cup T) = 0 = w_R(S \cup T')$ for all $S \subseteq N \setminus (T \cup T')$,

- $w_R(S \cup T) = 0 = w_R(S)$ for all $S \subseteq N \setminus T$ such that $S \cap T' \neq \varnothing$,

- $w_R(S \cup T') = 0 = w_R(S)$ for all $S \subseteq N \setminus T'$ such that $S \cap T \neq \varnothing$.

Hence by JSY $\phi_T(w_R) = \phi_{T'}(w_R)$. But $\phi_T(w_R) = p^T(R)$ and $\phi_{T'}(w_R) = p^T(R)$. This shows that $p^T(R) = p^{T'}(R)$ for all $\varnothing \neq R \subseteq N \setminus (T \cup T')$. To show that $p^T(\varnothing) = p^{T'}(\varnothing)$ we consider the game

$$w^*(S) = \begin{cases} 1 & \text{if } S \neq \varnothing, \\ 0 & \text{otherwise.} \end{cases}$$

Then

- $w^*(S \cup T) = 1 = w^*(S \cup T')$ for all $S \subseteq N \setminus (T \cup T')$,

- $w^*(S \cup T) = 1 = w^*(S)$ for all $S \subseteq N \setminus T$ such that $S \cap T' \neq \varnothing$ (since then $S \neq \varnothing$),

- $w^*(S \cup T') = 1 = w^*(S)$ for all $S \subseteq N \setminus T'$ such that $S \cap T \neq \varnothing$ (since then $S \neq \varnothing$).

It thus follows by JSY that $\phi_T(w^*) = \phi_{T'}(w^*)$. However, $\phi_T(w^*) = p^T(\varnothing)$ and $\phi_{T'}(w^*) = p^{T'}(\varnothing)$, which gives the required identity and shows that (6) holds.

**If:** Now we show the implication in the other direction. Suppose (6) holds and $v \in \mathcal{G}^N$ satisfies the three conditions in JSY. Then

$$\phi_T(v) = \sum_{S \subseteq N \setminus T} p^T(S)[v(S \cup T) - v(S)] = \sum_{S \subseteq N \setminus (T \cup T')} p^T(S)[v(S \cup T) - v(S)]$$

$$= \sum_{S \subseteq N \setminus (T \cup T')} p^{T'}(S)[v(S \cup T') - v(S)] = \sum_{S \subseteq N \setminus T'} p^{T'}(S)[v(S \cup T') - v(S)] = \phi_{T'}(v).$$

Hence JSY is satisfied. $\qquad\square$

*Proof of Theorem 1.* We have to show that there exists exactly one choice of constants $\{p^T(S)\}$ which satisfy equations (4)–(6). Notice that satisfying (5) and (6) is equivalent to satisfying

$$p^T(S) = p^{T'}(S) \, \forall \, S \subseteq N \setminus T, \, S' \subseteq N \setminus T' \text{ s.t. } s = s'.$$

Thus $p^T(S)$ does not depend on $T$ at all, and only depends on the cardinality of $S$. Let $q_s$ denote $p^T(S)$ for any $S \subseteq N \setminus T$. Then we can re-write equation (4) in terms of $q_s$ as

$$1 = \sum_{i=n-k}^{n-1} \binom{n}{i} q_i, \tag{10}$$

$$q_s = \frac{\sum_{i=(s-k)\vee 0}^{s-1} \binom{s}{i} q_i}{\sum_{i=1}^{k \wedge (n-s)} \binom{n-s}{i}} \quad \forall \, s \in \{1, \ldots, n-1\}. \tag{11}$$

Note that for any $q_0$, equation (11) fully determines all other $q_i$, for $i \in \{1, \ldots, n-1\}$ and $q_0$ is then determined by (10). Thus there is at most one solution. However, we have already identified (see the arrival-order discussion in Appendix A) that a solution to this recurrence is given by $(q_0, \ldots, q_{n-1}) = (\hat{p}_0, \ldots, \hat{p}_{n-1})$, for which $\hat{p}_0 = \left( \sum_{i=1}^{k} \binom{n}{i} \right)^{-1}$. $\qquad\square$

## C  STRONG JOINT SYMMETRY

We examine the effect of removing conditions 2 and 3 from JSY. As it turns out, this leads to the non-existence of an index. To be precise, we consider replacing axioms JAN and JSY with:

**SJS** *strong joint symmetry* : fix $\varnothing \neq T, T' \subseteq N$. Then

$$v(S \cup T) = v(S \cup T') \, \forall S \subseteq N \setminus (T \cup T')$$
$$\Rightarrow \phi_T(v) = \phi_{T'}(v).$$

**Proposition 5.** *There is no index $\phi$ satisfying axioms JLI, JNU, JEF, and SJS that is guaranteed to exist for all games.*

*Proof.* Since $\phi$ satisfies JLI, JNU, and JEF, by Proposition 2,

$$\phi_T(v) = \sum_{S \subseteq N \setminus T} p^T(S)[v(S \cup T) - v(S)]$$

with $\{p^T(S)\}$ satisfying (4), for any game $v \in \mathcal{G}^N$. We consider two games $v_1, v_2 \in \mathcal{G}^{\{1,2\}}$. As $N = \{1, 2\}$, (4) gives $p^{\{1\}}(\varnothing) = p^{\{2\}}(\{1\})$, $p^{\{2\}}(\varnothing) = p^{\{1\}}(\{2\})$, and $p^{\{1,2\}}(\varnothing) + p^{\{1\}}(\varnothing) + p^{\{2\}}(\varnothing) = 1$.

Suppose $v_1(\{1\}) = v_1(\{1, 2\}) = 1$, $v_1(\{2\}) = 0$. SJS thus gives that $\phi_{\{1\}}(v_1) = \phi_{\{1,2\}}(v_1)$, i.e. $p^{\{1,2\}}(\varnothing) = p^{\{1\}}(\varnothing) + p^{\{2\}}(\varnothing)$ which implies $p^{\{1,2\}}(\varnothing) = 1/2$.

Suppose also that $v_2(\{1\}) = v_2(\{1, 2\}) = v_2(\{2\}) = 1$. SJS gives that $\phi_{\{1\}}(v_2) = \phi_{\{2\}}(v_2) = \phi_{\{1,2\}}(v_2)$, i.e. $p^{\{1,2\}}(\varnothing) = p^{\{1\}}(\varnothing) = p^{\{2\}}(\varnothing)$ which implies $p^{\{1,2\}}(\varnothing) = 1/3$, giving a contradiction. $\square$

Thus, SJS is too strong a notion of symmetry, imposing linear restrictions on sets of unequal sizes.

