# OpenReview forum: "Joint Shapley values: a measure of joint feature importance"
_ICLR.cc/2022/Conference — ICLR 2022 Poster_

### Official Review · Reviewer_QfeR · 2021-10-31

**Correctness:** 3
**Technical Novelty And Significance:** 3
**Empirical Novelty And Significance:** 3
**Recommendation:** 8
**Confidence:** 2

**Main Review:**

Strengths. The paper is well written and the findings are well explained. The experimental section illustrates that the newly introduced joint Shapley values lead to reasonable results in real applications. Some examples of the joint Shapley values are provided in Table 5 to clarify the obtained results.

Weaknesses. The method is computationally heavy (but it is not easy to do it less computationally expensive). Indeed, the sampling can be done more efficiently, however, it is difficult to criticise this point much, since the authors mentioned it as their future work.


**Summary Of The Paper:**

The paper proposes an extension of the Shapley values, namely, the joint Shapley values which measure a set of features' average effect of a models prediction. The uniqueness of the joint Shapley values is proved. Moreover, training details and tuning parameters are provided in the accompanying code.

**Summary Of The Review:**

The topic is important and challenging. The results are novel, and the experimental section provides a nice illustration how the joint Shapley values can be used.

---

> ### Author Response · Authors · 2021-11-18
> **Thank you for your positive review**
>
> We agree that calculating exact joint Shapley value calculations is computationally intensive.  As mentioned, in passing, in the Conclusions, we are hopeful that efficient sampling techniques developed for classic Shapley values can be directly extended to joint Shapley values.  Indeed, as the joint Shapley is such a direct extension of the Shapley, we would be surprised if the Williamson & Feng (2020) and Mitchell et al. (2021) techniques were not easily extended.
>
> Please let us know if there is any further information we could provide to increase your confidence score?

---

### Official Review · Reviewer_Djge · 2021-11-01

**Correctness:** 4
**Technical Novelty And Significance:** 4
**Empirical Novelty And Significance:** 4
**Recommendation:** 8
**Confidence:** 3

**Main Review:**

I agree with the theoretical significance of this work, which generalizes the Shapley value for assigning a contribution of *a set of agents*.

Here are some comments:

1. In practice, how can we choose k? It would be great if the authors could provide some rules.

2. In the simulation, what is the explicit form of the set function? For example, in Table 2, even if there is an explicit form of a function f, it's nontrivial to derive the corresponding set function.

3. Continuing from 2, I recommend adding discussion on the choice of the set function.







**Summary Of The Paper:**

This paper introduces joint Shapley values, which extend Shapley values to measure contributions of *sets*.


**Summary Of The Review:**

- I agree with the theoretical significance of this work.

- A few questions/comments on the perspective of practitioners are listed above.

---

> ### Author Response · Authors · 2021-11-18
> **Thank you for your positive reading of our paper**
>
> In reply to your comments:
>
> 1. _choosing $k$_: We note, in the Introduction, the tradeoff between the additional insights gained by large $k$, at the price of higher computational costs.  We have added to the Conclusions a workflow that seems natural to us: compute $k = 1$ Shapley values, followed by $k = 2$ to identify strong pairwise effects, and then tailor $k \geq 3$ calculations in response to the analyst’s evolving questions about the model’s functioning.
> 2. *the set function*: we now number the equation linking the $f()$ 'ML model' to the set function, $v()$, on p.5.  Further, we cite it in the text introducing Table 2.  We note that this approach has been standard since Štrumbelj & Kononenko (2010).  For reasons of space, we do not develop this discussion: if you had specific concerns or questions about our current phrasing, we would be happy to incorporate them.
>
> Please let us know if we have missed anything, or if there is anything we can do to increase your confidence in your assessment.

---

### Official Review · Reviewer_VFxY · 2021-11-02

**Correctness:** 3
**Technical Novelty And Significance:** 3
**Empirical Novelty And Significance:** 2
**Recommendation:** 5
**Confidence:** 3

**Main Review:**

Strengths:
- The code looks quite good. Has walkthroughs, examples, etc. Implementation looks clean.

- The theoretical results look clean, intuitive, and well-presented.

- The work inlcudes some discussions on practical questions, including the employed hardware resources, and algorithm complexity estimates

- The work gives an overview of a good set of other SOTA approaches for Shapley




Weaknesses/Considerations:
- For XAI-based approaches, it is very important to consider their practical advantages, ease-of-use, and intuitiveness. Whilst this work gives a good treatment of the mathematical properties of the proposed approach (and related approaches), these properties are not necessarily tied to practical advantages. For example, emphasis is placed on the fact that "interaction indices are computed with discrete derivatives" (page 1), and yet, it is non-obvious what this implies from a practical standpoint.

- Furthermore, it feels as if this work is lacking a general summary of how to apply this approach for gaining insights on model behaviour. Currently, evaluation is done by running the approach on several values of "k", comparing the different outcomes, and tying it to properties, such as "negation effects". However, such an approach feels quite manually-intensive, involving significant efforts of post-hoc interpretation of the results. This will especially be the case with a larger number of features, and larger values of "k". Including a short description of how to apply this proposed approach in order to gain insights from a model with minimal manual effort involved would be helpful.

- Whilst this work lists numerous related SOTA approaches to feature importance (and Shapley values in particular), a comparison is only made on the toy benchmark datasets. Giving concrete examples of the differences in explanations provided by these approaches on the more complex cases (e.g. Boston Housing), would provide a much clearer view of their relative strengths and weaknesses. At the very least, a justification for why such a comparison is impossible would be helpful.

- There are quite a few explanation properties introduced in the Applications section, such as the "negation effect", or the "cancellation effect". If this is novel terminology introduced in this work, it would be helpful to discuss this earlier on in the methodology section (or in the appendices). If these properties are based on prior work - it would be helpful to include references. If not - giving more intuition on these properties (perhaps with more illustrative toy examples) would help.



Other Remarks:
- "evaluating it at a reference or baseline feature" (page 1) - I think there's a typo in that sentence

- The Introduction section is essentially merged with the Related Work section, making it a little hard to follow. I suggest splitting this into two separate sections for clarity.

- Is there any particular reason for why the "Applications" section is not called "Experiments"?

- Table 2: not sure if so many decimal places are required. Removing some of them could make the table more compact, and improve readability.


**Summary Of The Paper:**

This work introduces "joint Shapley values", which directly extend Shapley’s axioms and intuitions: joint Shapley values measure a set of features’ average effect on a model’s prediction. This work naturally extends Shapley's axioms from a single feature, to sets of features. In a nutshell: joint Shapley values measure the average marginal contribution of a set of features to a model’s predictions. This work presents rigorous mathematical results for the joint Shapley values approach. This approach is then evaluated on several datasets, including: (i) simulated data, (ii) Boston Housing data, (iii) Movie Review data.

**Summary Of The Review:**

The potential impact of this work is not clear as not enough is said about practical advantages.

Gaining some insights into the model behaviour would make the content of the paper stronger.

---

> ### Author Response · Authors · 2021-11-18
> **Thank you for your detailed and thoughtful reply**
>
> To address your comments:
>
> 1. *practical advantages of joint Shapley values*: we have edited the Abstract and Introduction to try to clarify the complementary roles played by the joint Shapley, as a measure of feature importance (providing insight into how knowledge of a set of features influences predictions), and interaction indices (providing insight into the internal dynamics of sets of features).
>    We have removed mention of the _discrete derivatives_ as we do not mention them elsewhere in the paper, so seemed a distraction.  The relationship of the joint Shapley value to an interaction index is similar to that between a [discrete] directional derivative and a [discrete] partial derivative: the directional derivative and joint Shapley values both provide direct measures of a change in a function's value in the specific direction of interest; they can be approximated by weighted sums of partial derivative and interaction indices, but - if one is interested in the function's change in a particular direction - the joint Shapley value provides a way of calculating it directly.  If a version of this explanation would clarify the paper, please let us know, and we will work to incorporate it.
> 2. *general summary for how to apply ... minimizing manual effort*: we have added comments to this effect to the Conclusion.  To unpack them a bit further: we are still gaining intuitions about how to use these techniques, so feel that we are not yet in position to give proper 'production' advice.
>    1. as noted in the new comments, we also see tools like the joint Shapley as interactive, as they are designed to enhance human understanding of the model.  Thus, we see them as similar to diagnostic tests in medicine: a doctor will prescribe tests as her understanding of a patient's condition evolves rather than according to a predetermined plan.
>    2. as noted in §4.2.1, 'enhancement and cancellation effects do not uniquely identify underlying phenomena'.  To us, this also points to the importance of manual effort in deriving intuitions: we feel that we are still a long way away from being able to algorithmically interpret the meaning of, say, a particular enhancement effect.
> 3. *comparison of measures on SOTA benchmarks*:
>    1. we have worked hard to cover a lot of ground concisely in the paper, to present an analysis with good coverage while still adhering to the page limit.  This has forced us to rely on some 'compression' techniques, such as trying to minimize the repetition of material already in the literature.  We work with the Boston and Rotten Tomatoes datasets as they are analyzed in Dhamdhere, Agarwal and Sundararajan (2020); thus, our analysis (for joint Shapley values) can be compared to their analysis (for Shapley-Taylor values).  Our analysis of the Boston housing data is able to slip in some explicit comparisons to the Shapley-Taylor, although mostly at the schematic level.
>    2. we have discussed, internally, more comprehensive comparisons of the joint Shapley value to existing measures.  Our concern is that doing so might confuse the fact that they seek to assess different phenomena, instead implying that they are comparable ways of measuring the same phenomenon.
> 4.  _negation effect ... cancellation effect_: in §4.2.3, we are clearer that we follow Dhamdhere et al. in applying intuitive names to the effects that we observe (e.g. 'negation effect').  In §4.2.1, we now explain that 'cancellation effect' is our term for the effect.  Given space restrictions, we do not think that we can present more toy models; if you felt that those presented were missing effects or interactions that might be of interest, please let us know: we should be able to replace one with a model that illustrated those effect better.
> 4.  *typo: 'evaluating it at a reference or baseline feature'*: we have reworded this for clarity.
> 4.  *Introduction merged with Related Work*: we agree; our rationale is, again, pragmatic: we are trying to squeeze as much content as possible into the page limit, so have suppressed a number of section and subsection heads, each of which buys us a few sentences.
> 4.  *Applications -> Experiments*: changed; thank you - this is more consistent with our seeing these analyses as preliminary.
> 4.  *Table 2's decimal places*: good point; clearer and more compact now.
>
> We hope that the revised manuscript and the above address your concerns, and allow you to raise your Recommendation score.  We hope also that you will let us know of any remaining concerns you might have, so we can seek to address them.

---

### Decision · Program_Chairs · 2022-01-20

**Decision:**

Accept (Poster)

**Comment:**

The reviewers think the topic is important and challenging. The results are novel, and the experimental section provides a nice illustration how the joint Shapley values can be used. However, the paper can be improved by including more real world applications and experiments.